# A Mechatronic Platform for Computer Aided Detection of Nodules in Anatomopathological Analyses via Stiffness and Ultrasound Measurements

**DOI:** 10.3390/s19112512

**Published:** 2019-05-31

**Authors:** Luca Massari, Andrea Bulletti, Sahana Prasanna, Marina Mazzoni, Francesco Frosini, Elena Vicari, Marcello Pantano, Fabio Staderini, Gastone Ciuti, Fabio Cianchi, Luca Messerini, Lorenzo Capineri, Arianna Menciassi, Calogero Maria Oddo

**Affiliations:** 1Sant’Anna School of Advanced Studies, The BioRobotics Institute, 56025 Pisa, Italy; sahana.prasanna@santannapisa.it (S.P.); elenavicari.ev@gmail.com (E.V.); marcello.pantano03@gmail.com (M.P.); gastone.ciuti@santannapisa.it (G.C.); arianna.menciassi@santannapisa.it (A.M.); 2Department of Information Engineering, Università Degli Studi di Firenze, 50139 Florence, Italy; andrea.bulletti@unifi.it (A.B.); m.mazzoni@ifac.cnr.it (M.M.); lorenzo.capineri@unifi.it (L.C.); 3Consiglio Nazionale delle Ricerche of Italy, Istituto di Fisica Applicata “Nello Carrara”, 50121 Florence, Italy; 4Azienda Ospedaliera Careggi University Hospital of Florence and University of Florence, 50134 Florence, Italy; francesco.frosini@unifi.it (F.F.); staderini.fabio@gmail.com (F.S.); fabio.cianchi@unifi.it (F.C.); luca.messerini@unifi.it (L.M.)

**Keywords:** cancer nodules detection, phantom, stiffness analysis, ultrasound analysis, visual analysis, automatic robotic platform, remote support for pathologists

## Abstract

This study presents a platform for ex-vivo detection of cancer nodules, addressing automation of medical diagnoses in surgery and associated histological analyses. The proposed approach takes advantage of the property of cancer to alter the mechanical and acoustical properties of tissues, because of changes in stiffness and density. A force sensor and an ultrasound probe were combined to detect such alterations during force-regulated indentations. To explore the specimens, regardless of their orientation and shape, a scanned area of the test sample was defined using shape recognition applying optical background subtraction to the images captured by a camera. The motorized platform was validated using seven phantom tissues, simulating the mechanical and acoustical properties of ex-vivo diseased tissues, including stiffer nodules that can be encountered in pathological conditions during histological analyses. Results demonstrated the platform’s ability to automatically explore and identify the inclusions in the phantom. Overall, the system was able to correctly identify up to 90.3% of the inclusions by means of stiffness in combination with ultrasound measurements, paving pathways towards robotic palpation during intraoperative examinations.

## 1. Introduction

Cancer is an abnormal and uncontrolled cell growth that invades healthy tissues, and that can spread via metastases to other locations in the body [1]. Various cancer treatments involve chemical and radiation therapies or surgery [2,3,4]. Following surgical intervention, biopsy is performed on the lymph nodes excised from the tissue to properly characterize cancer spread and examine whether it has developed the ability to spread to other lymph nodules or organs too. The accuracy in estimating the amount of spread of cancer is extremely important to avoid complications caused by an extensive resection of healthy lymph nodes and tissues. Accurate localization of tumors in tissues resected during surgery can also allow the surgeon to decide and modify in itinere the planned intervention so to remove malignant tissues missed in pre-operative imaging. Stiffness of human tissue is higher for tumor nodules with respect to healthy tissues [5,6,7,8,9,10]. Hence, inspecting the mechanical properties of cancerous tissues can contribute to the detection of nodules. Intraoperative palpations of the resected malignant tissue provide essential information about the presence of abnormalities [11]. Indeed, such investigation is part of the general practice performed by a specialist through manual palpation to retrieve several information about cancer nodules [12]. The reliable confidence of medical practitioners to detect tumors is achieved with rigorous training before they reach proper expertise in examining various organs and detecting abnormalities [13]. The human capability to detect lumps in the tissues, however, degrades with increasing lump depth, decreased compliance of the tissue, deformation of the finger pad induced by the lump itself, and the finger indentation velocity [14,15]. Ultrasound analysis [16] can complement stiffness data because of the different acoustic properties of cancer nodules, as demonstrated by intraoperative ultrasonography recordings having reported influence (varying from 2.7% up to 73%) on the surgical procedures that were preoperatively planned [17,18,19,20]. Elastography is also used for the investigation of many diseases in tissues (liver, breast, thyroid, kidney, prostate, and lymph nodes) [21]. The elastographic equipment is of two types depending on the US sensors arrangement: Strain imaging and shear wave imaging. In both cases a US array probe is gel coupled to tissues; the latter requirement represents a limiting factor for the anatomo-pathology examinations for which any contamination of tissue samples must be avoided and only the natural wetness of excised tissues is allowed.

In this study, we combined stiffness and ultrasound data to aid the intraoperative histological exams performed on tissues excised during surgery by the identification of regions with different stiffness and US impedance characteristics. Such an examination is crucial in case of misdiagnosis or in case of unforeseeable diagnostic queries that might arise during surgery. Results from the examination may be used as a guide for surgical resection and decision-making to modify the common surgical procedure (see Figure 1). Initially, the surgeon gets a small tissue specimen (2–3 cm) from patient in order to get information about the presence of pathological patterns. Then, this specimen is placed on an automated system able to move and scan the material in order to get the data. The system proposed in this work is based on a precision mechanical scanner that provides a map of the tissue sample with inclusions. Data were derived from a sensor designed with a load cell for stiffness measurements, which also ensures the contact force on the sample to a vertically supported needle-type ultrasonic transducer provided for US measurements. In this way, two types of data (mechanical stiffness and US impedance) are measured at the same time in each position providing in practice two maps with well-correlated information regarding the positions of inclusions within the scanned area. Finally, with the position of the inclusion the pathology lab technician is able to sample the material and create a precise number of slices for microscopy analysis. Directly from the operating theatre the images are sent to the pathologist for the remote diagnosis. The time between the initial part of the process (excision) and the final part (medical report) should be contained within 20–30 min.

With instrumented tools, automatic classification of tumors in tissues can be addressed by machine learning techniques: Supervised–unsupervised classification, clustering and learnt neural networks [22,23]. The proposed system aims at reproducing the activity of pathologists in intraoperative tumor identification using feedback from vision, stiffness [24] and ultrasound measurements [25]. Using a robotic platform, and machine learning techniques for classification, the focus of this work is to detect and localize nodules buried in phantoms mimicking the elastic and ultrasound properties of excised human tissues. Specifically, the experimental evaluation was carried out by means of Agar-based phantoms suited to mimic liver, cardiac, brain and soft tissues [26,27,28], either in their acoustic and mechanical properties and temperature dependency [29,30].

The paper is organized as follows. Section 2 describes the experimental setup, the technical specifications of the used phantoms, the experimental protocol and data analysis methods. Results are presented in Section 3, showing the results of stiffness and ultrasound data analyses both separately and merging them together. The last section concludes with the discussion of the entire work and presents potential future investigations.

## 2. Materials and Methods

### 2.1. Platform Design

A platform was developed to detect embedded rigid inclusions surrounded by a soft matrix. The automated system consists of the following components (Figure 2):(i)Three motorized translational stages and one rotational stage allowing to move the sample. A commercial stage (8MTF-102LS05, STANDA, Vilnius, Lithuania) with 10 cm of travel range and a resolution of 2.5 µm was used for the *X* and *Y* axes, while another translational stage (8MVT120-25-4247, STANDA, Vilnius, Lithuania) was used to indent the sample along the *Z* axis, having a travel range of 2.5 cm and a resolution of 5 µm. Additionally, a fourth stage was mounted on the mechatronic platform (8MR190-2-28, STANDA, Vilnius, Lithuania) in order to enable the rotation of the sample. Such stage had 360° rotation range with 0.01° resolution.(ii)An ultrasound probe (Sonomed, mod. 2014059, Warsaw, Poland) with 16 MHz central frequency and a fractional bandwidth equal to 0.25 at −6 dB used in pulse-echo mode. The needle-type probe, 3 mm in diameter, was selected for directly contacting and indenting the sample. A 30 Vpp pulsed excitation was delivered to the probe via a transmitter (US-Key, Lecoeur-Electronique, Chuelles, France) connected to a PC via USB2. The experimental setup was completed with the ultrasound data acquisition device, NI FlexRIO (National Instruments Corp., Austin, TX, USA), for acquisitions at high frequency (1.6 GHz).(iii)A load cell (Nano 43, ATI Industrial Automation, Apex, NC, USA) to collect interaction forces, up to 18 N with 0.004 N resolution along normal axis, arising at the interface between the ultrasound probe and the sample, also used in the control loop of the translation stages in order to operate force-controlled indentations. The developed software used this force data to calculate the stiffness and to trigger the high frequency US data collection at the threshold point of contact (0.2 N).(iv)A waterproof HD-camera (Hero5 Session, GoPro, San Mateo, CA, USA) with 10 MP and 4 K resolution, integrated to perform the sample shape recognition and to create a matrix of points to be indented.(v)A stainless-steel disk fixed on the top of the motorized stages for the positioning of the sample, but also to permit the reflection of the ultrasound signal back to the probe. The disk had a diameter of 16 cm and a thickness of 1 cm.

The software routines for controlling the platform and the automatic scan of the samples and for performing data acquisition, as well as the graphical user interfaces were developed in LabVIEW, LabVIEW Real-Time and LabVIEW FPGA (National Instruments Corp., Austin, TX, USA), while the data analyses were performed using MATLAB (The MathWorks, Inc., Natick, MA, USA).

### 2.2. Phantom of Healthy Tissue and Inclusions

Tests were performed on seven Agar block-shaped phantoms, realized to mimic both the mechanical and the acoustic properties of diseased human tissues. Each phantom had a soft surrounding matrix representing the human healthy tissue and hard inclusions embedded inside to represent tumor nodules. Each fabricated phantom was nominally 60 mm wide, 100 mm long and 15 mm thick, while the buried spherical inclusions had different diameters ranging from 3 mm to 12 mm. The volume of the phantom was large enough to introduce up to 8 inclusions, 2 per each diameter, in different *X*–*Y* positions with adequate separation distance (Figure 3) in order to execute computer-aided detection trials. It is worth to mention that the Agar phantom did not need any pre-treating before performing the automatic scan process. However, also in case of biological tissues, it is not necessary any further pre-treatments apart from the one required within custom histological evaluations.

Agar-based phantoms were prepared using a predefined concentration of Agar in distilled water. Changing the concentration of Agar resulted in a variation of both the mechanical and acoustic properties. A concentration of 2 g of Agar in 100 mL of water was used to represent a healthy human tissue (fabricating a phantom entirely with this concentration results in 1.59 MRayl acoustic impedance, 1457 m/s speed of sound and 0.33 N/mm mechanical impedance). A concentration of 8 g of Agar in 100 mL of water was used for simulating a tumor tissue (fabricating a phantom entirely with this concentration results in 1.92 MRayl acoustic impedance, 1534 m/s speed of sound and 4.6 N/mm mechanical impedance) [28,30,31,32].

### 2.3. Experimental Protocol

The experimental protocol consisted in an automatic scan of the sample. The procedure was divided in two steps:(i)Visual analysis;(ii)Stiffness and ultrasound analysis.

The purpose of the automatic visual analysis was to recognize the shape of the sample by acquiring its boundaries and to create the indentation matrix, namely the points to be analyzed. Such analysis is crucial when dealing with real tissues, where the shape and size is unknown or irregular, so that the scan can be defined automatically. The visual part (Figure 4) consisted in subtracting the background image from the sample image, thus obtaining the shape, the size and the orientation. Starting from this new image (Figure 4C), a set of indentation points was created with a 2 mm step along the *X* and *Y*-axes.

Once the visual analysis was completed, it was possible to start the acquisition of the compression force and ultrasound signals. Per each *X*–*Y* point of the indentation matrix, the phantom was indented along the *Z*-axis at constant speed (0.5 mm/s). The compression force was recorded and, at a low threshold (0.2 N, to avoid damaging the phantom), a trigger signal was generated for ultrasonic pulse transmission and reflected signal reception for recording (Figure 5). In a nutshell, the robotic platform control was fully automatic from the placement of the tissue onto the platform up to the localization of the inclusions. However, the system provided the user interface for an operator to supervise the scan according to the physician’s requirements.

### 2.4. Data Analysis

The detection and localization of the different inclusions was based on the elaboration of indentation force (*F_Z_*), vertical position (*Z*) and ultrasound signals. The stiffness parameter *k* is the ratio between the change in force and the change in Z displacement and for each indentation was calculated according to Equation (1). Here, $F_{z_{max}}$ refers to the force threshold of the indentation and $F_{z_{0}}$ refers to force measured by the load cell following the first contact.

(1)$$k=\frac{\Delta F_{z}}{\Delta Z}=\frac{F_{z_{max}}-F_{z_{0}}}{Z_{F_{z_{max}}}-\;Z_{F_{z_{0}}}}$$

The ultrasound technique used for the detection of the inclusions was based on the reflectometric method. In fact, we considered more reliable to work with the variation of the signal reflected from the interface created by the bottom of the phantom and the steel plate. This signal has a higher amplitude than the back scattered or reflected signal from the inclusion embedded into the tissue-like matrix. For our phantoms, according to the selected acoustic parameters mimicking healthy and cancerous tissues [31], we can estimate a reflection coefficient generated by the acoustic impedance difference at a planar interface less than 1%. The ultrasonic analysis consisted in the processing of the signal detected in each point of the indentation matrix defining a Correlation Index Amplitude (*CIA*) parameter derived from [33], as reported in Equation (2):(2)$$CIA=1-\left(\frac{min\left(\sqrt{\sum S_{ref}^{2}},\sqrt{\sum S_{i}^{2}}\right)}{max\left(\sqrt{\sum S_{ref}^{2}},\sqrt{\sum S_{i}^{2}}\right)}\right)$$

In Equation (2), *S_i_* is the signal acquired in each point and *S_ref_* is the reference signal. The *CIA* assumed values between 0 and 1: For two identical signals the CIA is zero, while for very different amplitude signals the *CIA* approaches to 1. We assume that a reference signal can be acquired in a position where neither inclusions nor other inhomogeneity have been detected; under this assumption the transmitted signal from the US probe is only attenuated by the two-way travel path defined by the steel plate interface. Due to low tissue sample thickness (typically from 5 mm to 20 mm) and moderate attenuation of the US at the operating frequency (1.25 dB/cm @16 MHz) this signal has an amplitude that is greater than the amplitude of the echo signals obtained in the probe positions over an inclusion; in our test samples, we found an amplitude variation of about 10% and 60% for the echo signal at the steel plate interface over and outside the inclusions, respectively. A high *CIA* indicates the detection of an inclusion since the two signals become poorly correlated.

For each indented point, a colour map was created both for stiffness and for correlation index amplitude. An unsupervised classifier, called Fuzzy C-mean (FCM) clustering [34], was used to classify each indentation of the scan on the phantom. Such unsupervised classification system, starting from the elaborated data, enabled the categorization of the point and the subsequent organization into different clusters. In this way, it was possible to divide the data into two classes: (a) Tumor class, which were the sites classified as inclusions, and (b) healthy class, which were the sites classified as non-inclusions. From the wrong classification prediction, we obtained the number of false positive, i.e., soft matrix points classified as inclusions, and the false negative number, i.e., inclusions classified as soft matrix. Furthermore, new datasets were obtained and classified by merging the stiffness and the ultrasound data using AND–OR logics. In the AND case, we considered tumor only the points identified as inclusion in both the datasets simultaneously, thus we expected an increase in the total number of false negatives. In the OR case, we considered tumor all the points classified as inclusion in either the stiffness dataset or the ultrasound dataset, thus we expected an increase of the number of false positives and reduced false negatives. The results of the OR logics are crucial to include all of the cancerous tissues. Through a confusion matrix, the accuracy and the misclassification rate were calculated for all the datasets and methods.

## 3. Results

All the experimental results presented in this section have been repeated over seven replicas of the developed phantoms.

### 3.1. Results from Stiffness Measurements

An elaboration example of the stiffness analysis, for one of the seven phantoms, is shown in the top parts of Figure 6. The bottom part of Figure 6A shows the positions of the inclusions inside the indentation matrix. Since the inclusions were embedded into a soft matrix, their stiffness was depending not only on the materials properties, like the elasticity, but also on their dimensions. The stiffness parameter recorded at the location of indentation is the complex homogenized combination of the inclusion below and the surrounding soft “healthy” tissue. Hence, the stiffness parameter, *k* from Equation (1), was found to increase with the dimension. Stiffness analysis was clearly capable to detect the bigger inclusions, namely 12 mm and 9 mm. Figure 6B, showing the results for the whole indentation matrix, confirmed this trend. A visual inspection of the image allows discriminating big inclusions compared to the soft surrounding matrix.

The results of the identification based on stiffness measurements are shown in Figure 7A, obtained by the Fuzzy C-mean (FCM) clustering. The results of this unsupervised classification system confirmed the ability of the stiffness measurement system to recognize all the points belonging to the big inclusions, thus without false negatives. Such performances were evident from the high number of true positive (green points) for 12 mm and 9 mm inclusions. However, stiffness analysis was not able to reliably identify the smallest inclusions, as pointed out by the high number of false negatives (red points) for 6 mm and 3 mm inclusions (Figure 7A).

### 3.2. Results from Ultrasound Measurements

According to the ultrasound data analysis, shown in Figure 6 (bottom part), we can observe in Figure 6A that the CIA index increases consistently in correspondence of the inclusions. However, unlike the stiffness measurements, higher CIA values were observed also for the smaller inclusions. Thanks to the high CIA peak recorded for each inclusion, this approach led to the detection of all the inclusions buried in the phantom. Figure 6B, showing the results for the whole indentation matrix, confirmed this trend. As for the stiffness measurement part, Figure 7B shows the results of the FCM clustering, highlighting the ability of the ultrasound system to detect each inclusion. The trend is visible in Figure 7B where true positives (in green) are present in each inclusion. Remarkably, false positives (in yellow) and false negatives (in red) were obtained in the area at the boundary between the inclusion and the soft matrix, confirming the high specificity in identifying the area to focus on for histological analyses. At such boundaries, the ultrasonic beam, coming from the source and returning to the source upon reflection on the stainless steel plate, could have experienced diffraction effects that produced an apparent enlargement of the real dimensions of the spherical inclusions thus giving origin to false positives.

### 3.3. AND–OR Logics to Merge Stiffness and Ultrasound Measurements

With the aim to improve the detection performance (true positives vs. false negatives), new datasets were obtained and classified by merging stiffness and ultrasound measurements using AND–OR logics and the corresponding results are shown in Figure 8. The AND logics (Figure 8A) turned out in an increase of false negatives and decrease of false positives. The growth of false negative predictions can lead to the worst-case scenario, since might bring to a loss of identified tumors. Instead, the OR logics demonstrated to be a safer approach since it turned out in an acceptable increase of false positives and a consistent decrease of false negatives. As shown in Figure 8B, the OR logics between stiffness and ultrasound measurements was able to correctly discriminate all the inclusions, even the smaller ones. Such results were achieved thanks to the complementarity of the two systems. It was observed that the stiffness analysis was better in localizing bigger inclusions, whereas the ultrasound analysis was better for the detection of smaller inclusions (compare Figure 7A,B).

This behavior was further confirmed by the confusion matrices obtained with the seven experimented phantoms and with all the identification techniques, i.e., based on just stiffness measurements, just ultrasound, and with the AND–OR logics (Figure 9).

## 4. Discussion

In this work we present a platform aiming at identifying cancer nodules in ex-vivo tissues. Such tool, oriented towards the automation of diagnostic procedures during surgery, has the scope of increasing the effectiveness of histopathological evaluations. Such exams need to be performed as correctly as possible because the report may lead in a modification of the surgical procedure. The human capability to detect these lumps with characteristic dimension of few mm, depends on the pathologist expertise and tactile capabilities. To achieve this goal, the presented platform combines three different measurements, such as camera vision, stiffness calculations via force-position sensing and ultrasound recordings to perform an automatic scan and evaluation of the indented tissue. In this paper the tests were performed in a laboratory environment using seven Agar phantoms that mimicked the mechanical and acoustic properties of human ex-vivo tissues. The phantoms integrated eight spherical inclusions with different diameters (from 3 mm up to 12 mm) to reproduce tumors inside healthy tissues. The results, for all phantoms, summarized in the confusion matrices, demonstrated the ability of the platform to automatically identify the inclusions, particularly when complementing stiffness with ultrasound measurements via OR logics. In particular, as reported in the confusion matrix, the tactile analysis presents valuable classification results in detecting the inclusions as reflected from the 78.73% of TP and 90.26% of TN. Moreover, it shows a low percentage of FP and FN, 9.74% and 21.27%, respectively. We observe that the tactile analysis provides satisfactory shape recognition and tumor detection for inclusions above 6 mm in diameter. On the other hand, it missed the smaller inclusions that were buried deeper into the softer matrix. The ultrasound analysis can be a very good guiding tool for localization and detection of tumors, including the smaller ones, because the ultrasound resolution is much higher than the size of the inclusion and the difference in the acoustic impedance along the *z*-axis is sufficient to generate an amplitude variation than can be detected from noise. The ultrasound data presents high amount of TN of 92.41% and a low FP of 7.59%. However, the ultrasound alone shows a high number of FN of 43.72%. To improve the performance, the classified datasets were logically merged using the OR and AND logics. As expected, the results of OR logics gave evidence of a higher rate of inclusions recognition (i.e., 90.3% of TP and 84.56% TN), while maintaining low error rates (i.e., 9.68% FN and 15.44% FP). Such a result is a direct consequence of the implementation of this logics, since we considered all the points classified as inclusion, in either the stiffness dataset or the ultrasound dataset, as tumor. This entails a better localization and reconstruction of the buried inclusions. Interestingly, the AND logics localizes the bigger inclusions with an increased TN rate of 98.10%and reduced the FP rate to 1.90%, but the TP rate of 44.70% and FN rate of 55.30% missed the correct shape and smaller tumors entirely.

In addition, we found that the ultrasound method was also sensitive to the presence of air bubbles formed in the agar inclusions as the surface of such bubbles might introduce a significant impedance contrast for the ultrasound signals. This could create amplitude variation in the reflected signal that resembled to the “healthy tissue”. Thanks to the good spatial resolution of our system, the positions around the air bubbles provide faithful data that can reduce the impact of this constraint; moreover, the biological tissues are expected to not have air bubbles. However, the tactile data were not sensitive to these air bubbles inside the inclusions, reproducing their shape a more faithfully in the OR logics. Within the present work, we adopted a scan resolution with step of 2 mm inspired by the 16 MHz needle probe diameter (i.e., 3 mm). To keep a balance between the scan speed and area, we decided to scan with step of 2 mm. Lesser resolution values lead to insufficient data points in the scanned area, while higher values would introduce unaffordable scan time and oversampling.

The phantoms we used were the simplistic versions of the biological tissues. Hence, further developments will address the experimentation of the robotic platform on ex-vivo tissues. After this validation step, we can envisage that the sensorized platform placed in the operating theatre will enable the pathologist to access data remotely with the purpose of assisting the surgeon in adapting the procedures during surgery. Information obtained from the platform can also be used to provide haptic feedback to the pathologist by means of wearable interfaces [35,36,37,38]. The analysis of vision data, now used only for detecting the boundary of the tissue and thus to define the indentation matrix, can be improved to provide a visual report too. Such a new procedure will target the extraction of several features from the pictures of both healthy and tumorous tissues to learn their differences via artificial intelligence methods and thus complement stiffness and ultrasound measurements. Finally, the results will be translated in an electronic report and integrated with the management software (e.g., HL7) of the healthcare system.

## Figures and Tables

**Figure 1 sensors-19-02512-f001:**
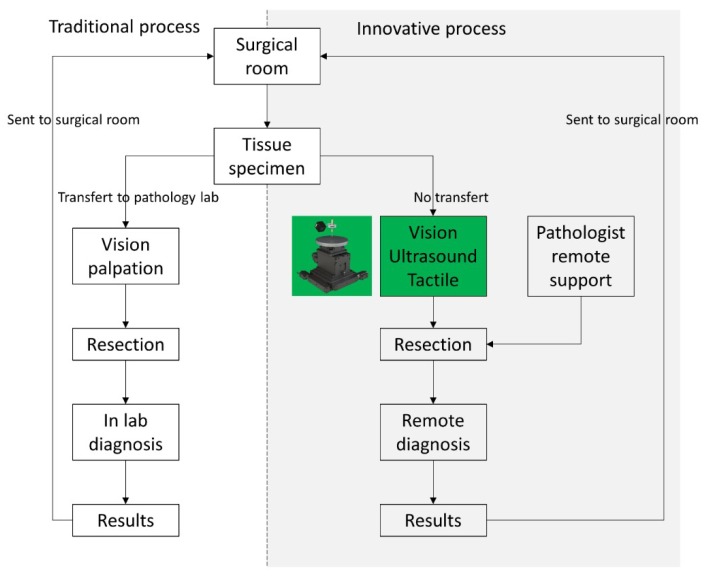
Block diagram of the histological procedure. On the left the traditional process is shown, whereas the proposed process is depicted on the right.

**Figure 2 sensors-19-02512-f002:**
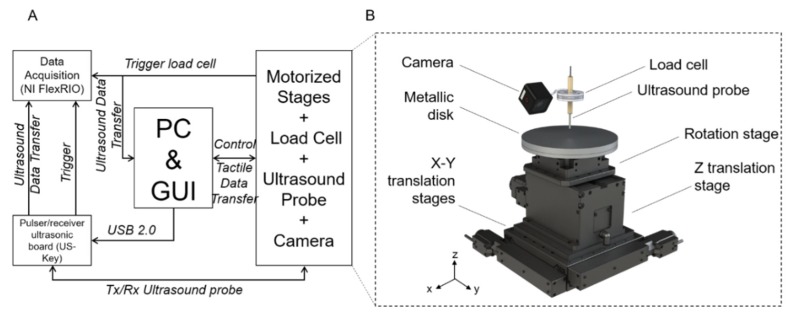
(**A**) Block diagram of the experimental setup. (**B**) Experimental setup showing the different components.

**Figure 3 sensors-19-02512-f003:**
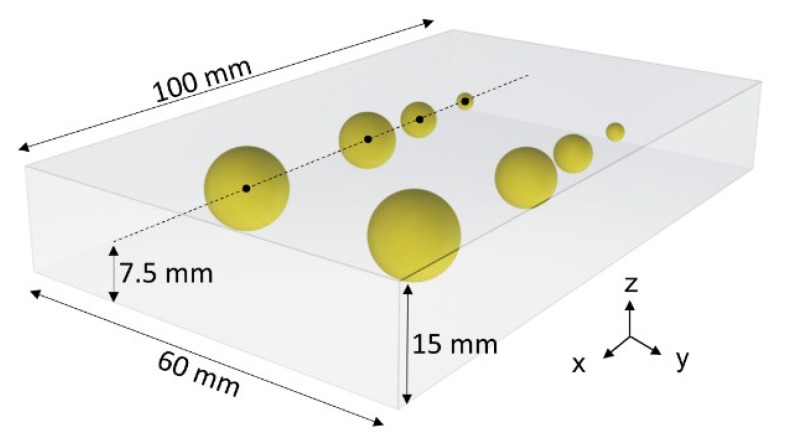
Rendering of the Agar phantom used during the experimental acquisition. The spherical inclusions are marked in yellow (∅ 12–9–6–3 mm). The volume of the phantom is 100 *×* 60 *×* 15 mm^3^.

**Figure 4 sensors-19-02512-f004:**
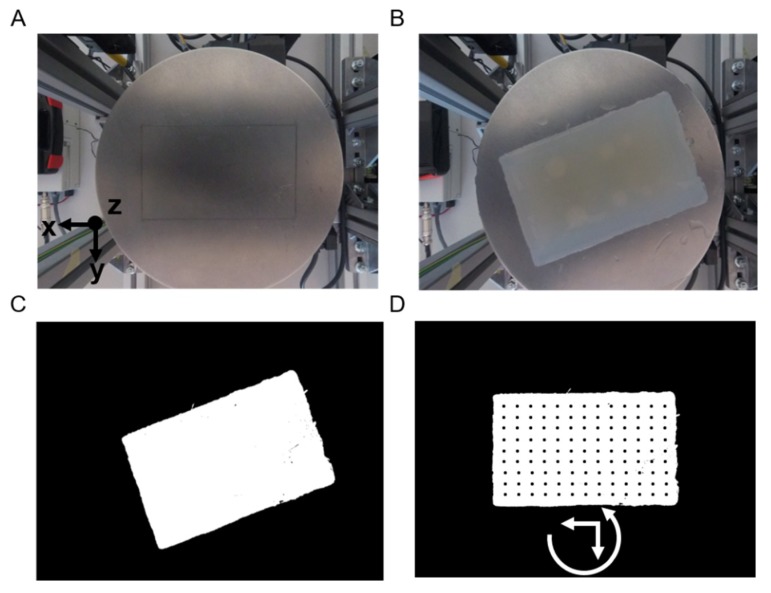
Visual part: Positioning of the sample, boundary detection and creation of the indentation matrix. (**A**) Background. (**B**) Sample in an arbitrary position. (**C**) Background subtraction. (**D**) Positioning by rotation of the sample and creation of the indentation matrix.

**Figure 5 sensors-19-02512-f005:**
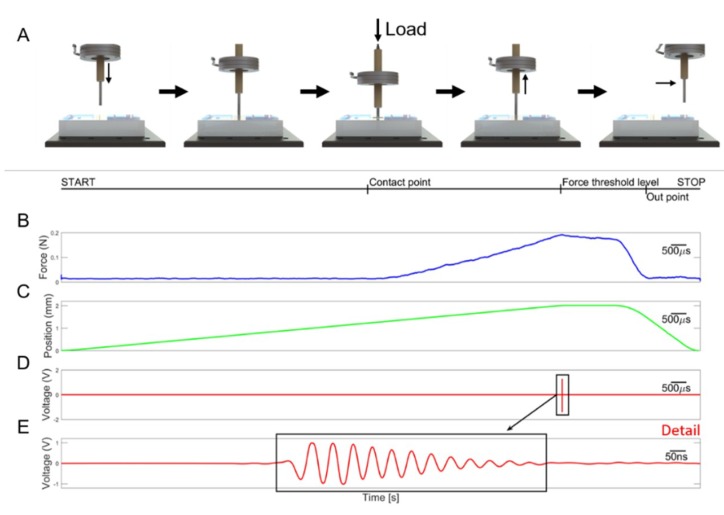
(**A**) Experimental protocol involving indentation of the ultrasound probe under regulation of the contact force. (**B**) Normal force. (**C**) Position along *Z*-axis. (**D**) Ultrasound signal reflected from the steel metal plate. (**E**) Zoom of the ultrasonic signal shown in panel D reflected at the tissue sample bottom in contact with the steel plate.

**Figure 6 sensors-19-02512-f006:**
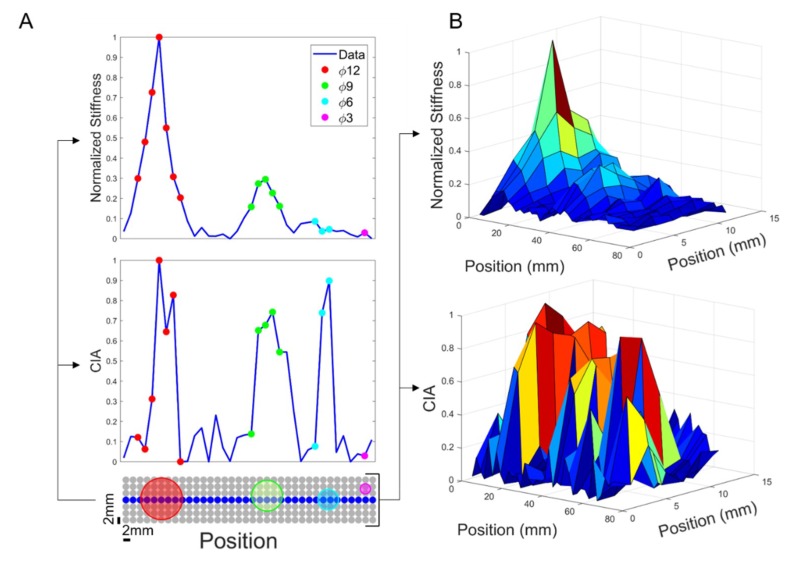
(**A**) (**top**) graph showing stiffness as a function of position, calculated as ΔFz/ΔZ, for the central row; (**bottom**) graph showing ultrasound signal processing of CIA index. (**B**) (**top**) 3D graph showing stiffness across the whole indentation matrix; (**bottom**) 3D graph showing ultrasound signal processing of Correlation Index Amplitude (CIA) index.

**Figure 7 sensors-19-02512-f007:**
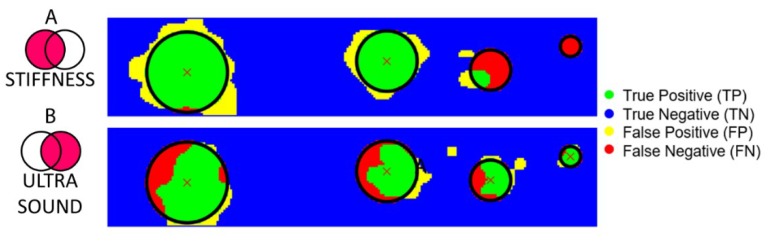
Classification (TP–TN–FP–FN) of all the points of the indentation matrix for the analyses with stiffness (**A**) and ultrasound (**B**) measurements.

**Figure 8 sensors-19-02512-f008:**
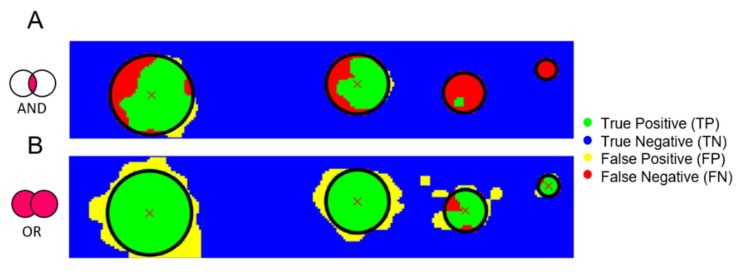
Classification (TP–TN–FP–FN) for all the points of the indentation matrix following the AND (**A**) and OR (**B**) logics of stiffness- and ultrasound-based classifications shown in Figure 7.

**Figure 9 sensors-19-02512-f009:**
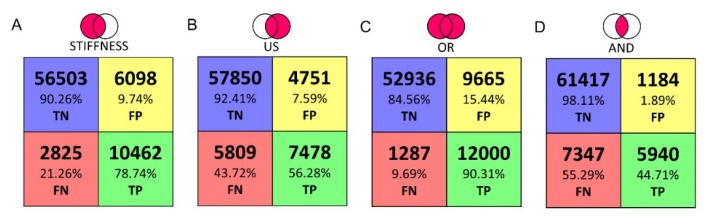
Confusion Matrix with classification based on (**A**) stiffness measurements; (**B**) ultrasound measurements; (**C**) stiffness or ultrasound measurements; (**D**) stiffness and ultrasound measurements.

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
