# Peer review of "A Mechatronic Platform for Computer Aided Detection of Nodules in Anatomopathological Analyses via Stiffness and Ultrasound Measurements"

_sensors, 2019, doi:10.3390/s19112512_

Round 1
Reviewer 1 Report
This manuscript proposed a platform for ex-vivo detection of cancer nodules, addressing automation of medical diagnoses in surgery and associated histological analyses. The system was able to correctly identify up to 90.3% of the inclusions by means of stiffness in combination with ultrasound measurements, paving pathway towards robotic palpation during intraoperative examinations. The paper is interesting and is recommended for publication. But there are some shortcomings in the paper, please revise them. How to pretreat tissue samples? The second is whether the control is fully taken into account in the design of the experiment. Line 283-284, it's a very confusing passage. Line 292-296, it needs more details to support the results of OR logic.Author Response
Authors: We thank the reviewer for the careful and positive evaluation of the paper, and for the requested amendments that help enhancing the scientific quality and clarity of our study. The manuscript has been corrected taking into account the indications provided by the reviewer in the decision letter. Please find below a point by point reply to the comments made by the reviewer.
Reviewer: Comments and Suggestions for Authors
This manuscript proposed a platform for ex-vivo detection of cancer nodules, addressing automation of medical diagnoses in surgery and associated histological analyses. The system was able to correctly identify up to 90.3% of the inclusions by means of stiffness in combination with ultrasound measurements, paving pathway towards robotic palpation during intraoperative examinations. The paper is interesting and is recommended for publication. But there are some shortcomings in the paper, please revise them.
Authors: We gratefully thank the reviewer for the positive consideration of our work. The following replies provide evidences of the revisions made in order to integrate the constructive amendments suggested by the reviewer to enhance the description of methods, contents and style.
Reviewer: How to pre-treat tissue samples?
Authors: In this study, we have validated the system using Agar phantom, which do not need pre-treating. However, also in case of biological tissues, our automatic indentation scans will not need for any further pre-treatments apart from the one performed within custom histological evaluations. Accordingly, we have updated the text as follows: “It is worth to mention that the Agar phantom didn’t need any pre-treating before performing the automatic scan process. However, also in case of biological tissues, it is not necessary any further pre-treatments apart from the one required within custom histological evaluations.”.
Reviewer: The second is whether the control is fully taken into account in the design of the experiment.
Authors: We thank the reviewer for the useful comment. The control of the robotic platform is automatic from the point where the tissue is placed on the platform up to the localisation of the inclusions. However, it was possible for an operator to supervise the scan according to the physician requirements (i.e., scan area, force threshold). The text has been updated as follows: “In a nutshell, the robotic platform control was fully automatic from the placement of the tissue onto the platform up to the localization of the inclusions. However, the system provided the user interface for an operator to supervise the scan according to the physician’s requirements.”.
Reviewer: Line 283-284, it's a very confusing passage.
Authors: We thank you for the comment. We agree that “up to 6mm” gave an impression of <6mm, while we expected >6mm. Please find the correction hereafter: “We observe that the tactile analysis provides satisfactory shape recognition and tumor detection for inclusions above 6 mm in diameter.”.
Reviewer: Line 292-296, it needs more details to support the results of OR logic.
Authors: According to the reviewer’s comment we frame better the results of OR logics by adding some details. The text has been changed as follows: “As expected, the results of OR logics gave evidence of a higher rate of inclusions recognition (i.e. 90.3% of TP and 84.56% TN), while maintaining low error rates (i.e. 9.68% FN and 15.44% FP). Such a result is a direct consequence of the implementation of this logics, since we considered all the points classified as inclusion, in either the stiffness dataset or the ultrasound dataset, as tumor. This entails a better localization and reconstruction of the buried inclusions.”.
Reviewer 2 Report
This work presents a platform for detection of tumor tissues based on the analysis of both stiffness and ultrasound in a phantom with exvivo measurements. Data are organized into different clusters to provide useful and reliable information. The work includes the use and control of a complex experimental setup and several mathematical tools. The work is very well written and authors transmit the interest and perspective of their work in a simple manner. I enjoyed much reading it. I think it is an interesting contribution to the field.
I include below some comments related to minor aspects of their work. I hope authors appreciate to considered them to improve the comprehension of some technical aspects. I recommend this work for publication in Sensors.
- The bibliography includes relevant references about ultrasonography and also medical analysis based on stiffness. However, I missed some references to elastography. That is, the techniques based on medical imaging mapping the elastic properties and stiffness of soft tissue. Ultrasound and magnetic resonance imaging (MRI) are both well known. Maybe in this work reference to the ultrasound elastography are more relevant (than MRI) as ultrasound is also used for tumor tissue detection. Indeed, authors may consider to use in the future only one sensor based on ultrasound for both phenomena (sonography and elasticity). They may also explain in the paper why not doing so (simplicity, disponibility, accuracy,…?)
- Figure 1 is conceptually complex. May details are out of the context of the written work (sampling, operating theater, …). I’m not sure that readers that are not familiarized with histological procedures can understand it easily. The explanation in the text is very simple at does not help. I propose the authors one of these options to correct this: 1) simplify it and explain the most relevant aspects as regard to the work or 2) include a paragraph explaining (chronologically or logically) the idea of the procedure.
- Figure 3 includes a nice qualitative representation of the sample. But the cartesian coordinate system is completely useless as no dimension has been included. Readers may appreciate if additionally, relevant geometrical information in one or two views XZ (at least the depth and position of the center of one line of spheres).
- Line 143. Acoustic relevant information about the simulated tumor tissue are included. A reference to justify these values based on real tissues may be important to justify the election.
- Eq. 1. and figure 5. Time dependent magnitudes are shown in figure 5 (force and position). From these magnitudes a single value of k has been obtained. How is this procedure? Is k a time averaged value obtained from measurements?
- Authors explain in line 178 that they consider “the variation of the signal reflected from the reference steel plate”. Please clarify what are the signals that have been correlated. What setups are used and (if necessary) adding an inset in Fig. 5 with a simple scheme of both ultrasonic signal configurations.
- Line 191, consider including a reference for the fuzy logic classifier.
- Lines 211 and 212. Interestingly authors point out that the stiffness does not depend only on the material, but also on the size of the inclusion. I have checked and authors never refer k as the elasticity of the inclusions, but usually in the literature it is the case. This is relevant in their work as the size of the inclusion is a significant parameter for detection. This is why I suggest them to expand on their clarification by explaining that k, the stiffness parameter, is not the stiffness of the tumor but a parameter that homogenizes the stiffness of the tumor and the surrounding “health” tissue.
- Fig. 7 and 9. In coherence with the word used in the text, “tactile” may be replaced by “stiffness”. Although it is clear reading the text, the word “Tactile” may induce confusion in fast readers as they may think it is not a stiffness sensor that has been used, but direct human inspection.
- Line 241. Results with ultrasounds are poor in the area at the boundary between the inclusion and the soft matrix. Diffraction of ultrasonic waves at the edges of the inclusion may be the reason for this flurry information.
- Line 254. Area of the detected inclusion is highly related with both stiffness and ultrasound analysis. Maybe authors have already considered the area as a relevant parameter to include in the decision algorithm by weighting both analyses. If it is the case, some comments on perspectives in this direction may be appreciated by the reader.
- Line 297. Air bubbles is sensitive to bubbles as the impedance contrast is significant and reflections of waves are produced.
Author Response
Dear Reviewer,
attached, please find our reply letter.
Thank you very much again for your constructive comments and best regards,
Calogero M. Oddo

Round 2
Reviewer 2 Report
Authors have considered all suggestions, answered them and changed the text accordingly. I confirm my positive recommendation for publication of the work in the journal Sensors-